# Suppression of Grape White Rot Caused by *Coniella vitis* Using the Potential Biocontrol Agent *Bacillus velezensis* GSBZ09

**DOI:** 10.3390/pathogens11020248

**Published:** 2022-02-14

**Authors:** Xiangtian Yin, Tinggang Li, Xilong Jiang, Xiaoning Tang, Jiakui Zhang, Lifang Yuan, Yanfeng Wei

**Affiliations:** Shandong Academy of Grape, Shandong Academy of Agricultural Sciences, Jinan 250100, China; yinxiangtian@shandong.cn (X.Y.); litinggang@shandong.cn (T.L.); jiangxilong@shandong.cn (X.J.); tangxiaoning@shandong.cn (X.T.); Zhangjiakui@shandong.cn (J.Z.)

**Keywords:** biocontrol agent, *Bacillus velezensis*, antagonistic activity, grape white rot, *Coniella vitis*

## Abstract

Grape white rot caused by *Coniella vitis* is prevalent in almost all grapevines worldwide and results in a yield loss of 10–20% annually. *Bacillus velezensis* is a reputable plant growth-promoting bacterial. Strain GSBZ09 was isolated from grapevine cv. Red Globe (*Vitis vinifera*) and identified as *B. velezensis* according to morphological, physiological, biochemical characteristics and a multilocus gene sequence analysis (MLSA) based on six housekeeping genes (16S rRNA, *gyrB*, *rpoD*, *atpD*, *rho* and *pgk*). *B. velezensis* GSBZ09 was screened for antifungal activity against *C. vitis* under in vitro and in vivo conditions. GSBZ09 presented broad spectrum antifungal activity and produced many extracellular enzymes that remarkably inhibited the mycelial growth and spore germination of *C. vitis*. Furthermore, GSBZ09 had a high capacity for indole-3-acetic acid (IAA) production, siderophore production, and mineral phosphate solubilization. Pot experiments showed that the application of GSBZ09 significantly decreased the disease index of the grape white rot, directly promoted the growth of grapes, and upregulated defense-related enzymes. Overall, the features of *B. velezensis* GSBZ09 make it a potential strain for application as a biological control agent against *C. vitis*.

## 1. Introduction

Grape white rot (hail disease) is prevalent in almost all grapevines worldwide, and it severely affects grape growth and results in a yield loss of 10–20% annually [1,2]. *Coniella vitis* is considered to be the main pathogen of the disease in China [3], although white rot can also be caused by *C. doplodiella* (Speg.) petr. and Syd. [4], *C. fragariae* (Oudem) B. Sutton, and *Pilidiella cataneicola* [5]. *C. vitis* mainly infects ripening berries, but also damages the green tissues of the vine [6]. Infected berries finally became soft and rotten and covered with pycnidia of grayish–white color. In addition, this seriously affects the quality and nutrition of grapes, with a production loss of 16.3% in China [3].

Plant growth-promoting rhizobacteria (PGPR) are bacteria that can promote plant growth. The mechanisms of PGPR mainly include the production of indole-3-acetic acid (IAA), fixation of nitrogen, dissolution of phosphorus, dissolution of potassium, secretion of ferriphagin, and secretion of plant hormones and related regulatory substances [7,8]. The ability of IAA biosynthesis in *Bacillus* varies depending on the species [9,10]. *B. velezensis* group strains are reputed to be PGPR. The production of IAA and cytokinin, both of which are associated with plant growth promotion, were also observed in *B. velezensis* [11,12]. PGPR have the ability to solubilize inorganic and/or organic phosphate from soil. *B. velezensis* ZF2 was identified as a plant growth-promoting biocontrol agent with the ability to produce IAA and solubilize mineral phosphate [13].

To date, 19 species of *Bacillus* have been isolated and identified, including *B. cereus*, *B. thuringiensis*, *B. subtilis*, *B. anthracis*, *B. atrophaeus*, and *B. amyloliquefacies* [7]. Previous studies demonstrated that *B. velezensis* was a heterotypic synonym of *B. amyloliquefaciens* based on DNA-DNA relatedness values [14]. Currently, *B. velezensis* can be distinguished from *B. amyloliquefaciens* and *B. subtilis* based on multilocus sequence analysis (MLSA) [13]. *B. velezensis* can grow under different environmental conditions with a wide range of temperature adaptations (15–45 °C) and exhibits tolerance to strong acids or strong alkalinity (pH 5–10) [15]. In addition, these species can grow in high salt concentrations of up to 12% NaCl and 2 g/kg NaCl in soil and is resistant to many antibiotics, including amoxicillin, ampicillin, rifampicin, fosfomycin, cefotaxime, cephalothin, piperacillin, streptomycin, sulfamethoxazole, ticarcillin, and trimethoprim [16,17].

As a member of the PGPR family, *Bacillus* has been widely used in disease prevention and control for crops and animals [18,19]. *B.*
*altitudinis* GLB194 and *B. pumilus* GLB197 have strong inhibitory activities against grape downy mildew [20,21], and *B. cereus* NRKT can increase the content of resveratrol in grapes and further reduce the occurrence of grape anthracnose [22]. *B. velezensis* was first described in southern Spain in 2005 [23], and recently, it was considered a potential biocontrol agent for controlling plant diseases due to its ability to stably colonize the rhizosphere of plants or the soil environment and provide nutrients for plant development. *B. velezensis* can produce fungal cell wall-degrading enzymes, which is an important mechanism for inhibiting pathogens [11]. Moreover, gene clusters for the biosynthesis of secondary metabolites were identified in *B.* velezensis, including surfactin, iturin, bacillomycin, fengycin, mycosubtilin, bacillaene, and bacilysin, most of which exhibit a wide range of antimicrobial activities [24]. *B. velezensis* strain KOF112 was isolated from the xylem of grape buds, showed good antifungal activity against *Botrytis cinerea*, *Colletotrichum gloeosporioides* and *Plasmopara viticola*, and significantly upregulated the expression of genes encoding class IV chitinases and β-1,3 glucanase in grape leaves [25]. *B. velezensis* Bvel1 produces various metabolites, including iturin A2, surfactin-C13 and C15, oxydifficidin, bacillibactin, l-dihydro anticapsin and azelaic acid [26].

Grape white rot caused by *C. vitis* has significantly affected the grape industry. Biological control is considered safer and more non-polluting than chemical prevention, considering grapes are usually used as a fresh fruit and for wine production. To obtain an effective strain for exploiting microbial agents to control grape white rot, in this study, *B. velezensis* strain GSBZ09, which has antagonistic activities against a broad range of fungal and bacterial pathogens in grapes, was isolated from the rhizosphere soil of grapevines. The biocontrol effect on grape white rot, antifungal mechanism and the plant growth-promoting ability of GSBZ09 were evaluated under both in vitro and in vivo conditions.

## 2. Results

### 2.1. Biocontrol Effect of B. velezensis GSBZ09 against Plant Pathogens

Fifty strains of *Bacillus* were isolated from rhizosphere soil of vineyards, of which nine strains exhibited antifungal activity against *C. vitis* and had a mycelial growth inhibition rate higher than 50% (Appendix A). Among these nine strains, GSBZ09 had the strongest antifungal activity against *C. vitis*, with a mycelial growth inhibition rate of 64.44% (Figure 1A). The mycelial growth inhibition rates of culture filtrates at 20% and 1% were 95% and 42%, respectively (Appendix A). The marginal hyphae inhibited by GSBZ09 had a significantly thick and curved morphology, showed more branches and exhibited tips that expanded into spheres, with the protoplasm flowing out (Appendix A). Spore germination of *C. vitis* was significantly inhibited by dilution of the culture filtrate with an inhibition ratio of 90%. Microscopic observations revealed that the germ tube could not elongate, both the center of the germ tube and the spores swelled, and the contents leaked out (Appendix A). In addition, strain GSBZ09 showed a broad antifungal and antibacterial spectrum, including activity against *Gloeosporium fructigrum*, *B. cinerea*, *Diaporthe eres*, *Alternaria viticola*, *Fusarium oxysporum*, *Aspergillus niger*, *Pestalotiopsis clavispora*, and *Allorhizobium vitis*. The inhibition rates on *G. fructigrum*, *B. cinerea*, and *P. clavispora* were 62.58%, 68.15% and 71.85%, respectively (Figure 1B).

### 2.2. GSBZ09 Was Identified as Bacillus velezensis

GSBZ09 was determined to be a Gram-positive, endospore-forming, and aerobic bacterium belonging to the *Bacillus* family. The strain grew well on lysogeny broth (LB) plates at 28 °C and produced creamy white colonies with irregular margins after 48 h of incubation, and they showed a colony morphology and culture characteristics similar to those of other *Bacillus* strains (Figure 2A). Strain GSBZ09 displayed rod-shaped cells with a length of 3–5 μm and a diameter of 0.8–1.2 μm (Figure 2B,C). The strain could grow in 2.0–6.5% NaCl and over a wide temperature range (15–80 °C), and the optimal pH was 7.0. GSBZ09 showed an exponential growth stage from 4 h post-inoculation (hpi) to 24 hpi. Importantly, we also observed that the pH value increased to 8.18 from an initial value of 7.0 during the culture time (Figure 2D). Biolog assays showed that strain GSBZ09 could utilize diverse carbon sources (Appendix A). To understand the genetic relationships between strain GSBZ09 and other *Bacillus* strains, a phylogenetic tree was constructed based on six housekeeping genes (16S rRNA, *rpoD*, *pgk*, *gyrB*, *atpD*, and *rho*) (Appendix A). As expected, four primary monophyletic clades, *Paenibacillus* sp. *B. subtilis*, *B. amyloliquefaciens*, and *B. velezensis*, were corroborated by bootstrap values. The phylogenetic tree showed that strain GSBZ09 was closest to *B. velezensis* FZB42. Based on these data, GSBZ09 belongs to *B. velezensis* (Figure 2E).

### 2.3. Detection of Extracellular Enzyme Production and Growth-Promoting Traits of Strain GSBZ09

The potential of strain GSBZ09 to antagonize fungi and promote plant growth was evaluated by characterizing several indicators on the plates. Extracellular enzyme assays showed that strain GSBZ09 produced cellulase, protease, amylase and lipase, thus indicating antagonistic traits against fungi. GSBZ09 also has the potential to decompose inorganic phosphorus to fix nitrogen. This strain can produce IAA and siderophores, which indicate its high potential as a competitive strain for biological control (Figure 3).

### 2.4. Antibiotic Resistance and Hemolysis Assay

The minimum inhibitory concentration (MIC) is defined as the lowest concentration of antimicrobial agent that inhibits the growth of a microorganism, and the minimum bactericidal concentration (MBC) is the lowest concentration of disinfectant lethal to the bacterium. In this study, strain GSBZ09 exhibited resistance to spectinomycin and showed a MIC of spectinomycin of 216 μg/mL and MBC of 1024 μg/mL (Figure 4). For spectinomycin, strain GSBZ09 showed no tolerance to the antibiotic (MBC/MIC = 4.74). A blood agar hemolysis assay confirmed that strain GSBZ09 was unable to produce hemolysin activity on plates (Appendix A).

### 2.5. GSBZ09 Has High Biocontrol Efficiency on Grape White Rots Caused by C. vitis

The biocontrol efficacy of strain GSBZ09 against *C. vitis* was evaluated using the leaves and fruit of *V. vinifera* cv. Red Globe (RG). The leaves and fruits of RG inoculated with only *C. vitis* were used as a positive control, and lesions on the leaves and fruits were observed at 3 dpi (days post-inoculation), with an incidence (disease index) of 72.73% (28.95) and 90% (82.96) in the fruit and leaves, respectively. The negative control (LB medium, GSBZ09 culture, and culture filtrate) exhibited no lesions on the leaves or fruits. To analyze the effect of GSBZ09 on preventing and controlling grape white rot, two treatments were conducted. One treatment consisted of grape inoculation with GSBZ09 24 h before inoculation with *C. vitis* to determine the preventive effect, and the other was inoculation with GSBZ09 24 h after inoculation with *C. vitis* to determine the control effect. The results showed that the control efficiency of the GSBZ09 culture (culture filtrate) in preventing the disease was better than that of the control, and the GSBZ09 culture exhibited higher biological control efficacy than culture filtrate (Figure 5). For the prevention treatment, the incidence and disease index of fruits treated with the culture (culture filtrate) decreased by 56.94% (52.73%) and 26.03% (23.02), respectively, while the incidence and disease index of leaves treated with the culture (culture filtrate) decreased by 63.33% (53.33%) and 78.51% (70.37), respectively.

### 2.6. Effects of GSBZ09 and the Culture Filtrate on Antioxidant Activity and Plant Growth Promotion

The activities of three protective enzymes, namely superoxide dismutase (SOD), polyphenol oxidase (PPO) and phenylalanine ammonia (PAL), increased to different degrees after treatment with GSBZ09 culture and culture filtrate. Compared with the control, the activities of the three protective enzymes increased 35.21% (SOD), 24.15% (PPO) and 0.62% (PAL) after irrigation with GSBZ09 culture. Meanwhile, the activities of the three protective enzymes increased by 22.52% (SOD), 20.70% (PPO) and 26.84% (PAL) after irrigation with the culture filtrate (Figure 6). In addition, a significant increase in the activities of the protein and proline levels in the *V. vinifera* leaves of the cv. RG and a decline in malondialdehyde (MDA) in the leaves after irrigation with GSBZ09 culture and culture filtrate. The results of the greenhouse experiment revealed that irrigation with GSBZ09 culture and culture filtrate significantly promoted the length (weight) of the aboveground parts and roots compared with the negative control. Obvious differences were not observed between the culture and culture filtrate treatments. The rates of growth promotion for the length, fresh weight and dry weight of shoots (roots) were 30.99% (23.85%), 98.69% (17.90%) and 55.18% (30.82%), respectively (Figure 6).

## 3. Discussion

Grape white rot caused by *C. vitis* has been reported as one of the main grape fungal diseases in China and results in an annual production loss of 16.3% [2]. To date, few studies have focused on the biocontrol of grape white rot disease. *Bacillus* spp. exhibit broad-spectrum biological activities against various phytopathogens [7,27], and *B. velezensis* has been reported to have good antifungal activity against fungal pathogens on grapes, such as *B. cinerea*, *C. gloeosporioides* and *P. viticola* [25,28]. In this study, nine antagonistic *Bacillus* isolates out of 50 rhizobacterial isolates were screened with *C. vitis* as targets. The nine antagonistic bacterial isolates exerted various levels of activity against *C. vitis*, and *B. velezensis* GSBZ09 showed the strongest antagonistic activity, with wide-spectrum biological activities against *G. fructigrum*, *B. cinerea*, *D. eres*, *A. viticola*, *F. oxysporum*, *A. niger*, and *P. clavispora*. Furthermore, the culture filtrate of GSBZ09 can also inhibit the mycelial growth of *C. vitis*.

Many PGPR have been reported to enhance plant growth through a wide variety of mechanisms. IAA is an important phytohormone that controls cell enlargement and tissue differentiation in plants, and many PGPR bacteria, including *Bacillus*, *Pseudomonas*, *Agrobacterium*, *Klebsiella* and *Rhizobium*, can produce IAA to stimulate the growth of plants [7]. *B. thuringiensis* RZ2MS9 was reported to promote tomato growth by the production of IAA [29]. In our study, a high IAA biosynthetic capacity (25.56 μg/mL) was exhibited in strain GSBZ09, and the pot experiment also demonstrated its effect on plant growth promotion. Phosphate is an essential and necessary element for plant growth and development, and mineral or organic forms of phosphorous are mainly found in soil; however, both forms are unavailable to plants [30]. GSBZ09 has been shown to be a phosphate-solubilizing bacterium that contributes to the utilization of P from the soil by plants. Siderophores have an exceptionally high affinity for Fe^3+^ and are able to bind Fe-siderophore complexes, thus promoting Fe uptake by microorganisms, while complexes can also be used by plants to increase the iron content inside plant tissues and improve plant growth [31]. In this study, GSBZ09 was observed to have a higher ability to produce siderophores. Overall, our results suggested the potential of GSBZ09 for improving plant growth.

Antibiotic resistance was established with secular evolution, which contributes to strong environmental adaptability [32]. Tolerance is defined as the inhibition of bacterial growth without lysis, with MBC values at least 32-times the MIC values [33,34]. The strain *B. velezensis* GSBZ09 exhibited resistance to spectinomycin but not resistance to ampicillin, vancomycin, kanamycin, streptomycin, gentamicin, chloramphenicol, tetracycline, or rifampin in our research. For spectinomycin, GSBZ09 showed no tolerance to the antibiotic (MBC/MIC = 4.74). Moreover, to use *B. velezensis* strains as effective biocontrol agent, they must be safe for animals, humans and the environment; however, the hemolysis phenotype is an indicator of toxicity. Previous studies reported that the biocontrol agents *B. cereus* and *Rahnella aquatilis* ZF7 showed hemolysin activity on plates [35]. Compared to *B. cereus* and *R. aquatilis* ZF7, *B. velezensis* GSBZ09 was unable to produce hemolysin activity, which indicated that it is friendlier to biological systems.

The biocontrol of plant pathogens by *Bacillus* occurs with the production of fungal cell wall-degrading enzymes, which is an important mechanism for inhibiting pathogens. *B. velezensis* ZF2 has been reported to produce protease (Prt) and cellulase (Cel), which may inhibit the mycelial growth of *C. cassiicola* and suppress cucumber leaf spot disease [9]. The current study demonstrated that *B. amyloliquefaciens* strain XZ34-1, as a biocontrol agent for the control of *Bipolaris sorokiniana*, has the ability to produce protease and pectinase [36]. In our study, GSBZ09 produced cellulase, protease, amylase and lipase, which may have resulted in the observed morphological changes and growth inhibition of *C. vitis* hyphae. Additionally, GSBZ09 culture and culture filtrate exhibited very good prevention and control effects on grape white rot caused by *C. vitis*. Biotropic bacteria induce systemic acquired resistance (SAR) in plants, which provides enhanced resistance to incoming pathogens [37,38]. Changes in defense enzymes (PAL, PPO, SOD and CAT) are usually used as important indicators to measure the defense response in plants [39]. MDA can affect plant growth and development by damaging the cell membrane [40]. In our study, the MDA content was reduced, suggesting less cell damage. The content of defense enzymes and proline increased in grape leaves treated with *B. velezensis* GSBZ09, which indicated that GSBZ09 increased the resistance of the grape by removing the reactive oxygen clusters. All these features suggested that strain GSBZ09 could be a promising biocontrol agent for grape white rot disease.

## 4. Materials and Methods

### 4.1. Isolation of Bacillus Strains and Antagonism Assays

*B. velezensis* GSBZ09 was isolated from soil around the roots of a healthy grape plants in vineyards in Jinan city, Shandong Province, China, in July 2018. Ten grams of soil were weighed and added to a triangular flask containing 90 mL of sterilized saline, followed by a water bath at 80 °C for 20 min, 10,000-fold dilution, plating on an LB plate (5 g yeast extract, 10 g peptone, 10 g NaCl, 15 g agar, 1000 mL distilled water, pH 7), and incubation at 28 °C for 24 h. *Bacillus*-like strains were selected based on the colony morphology, and used for further screening of antagonistic bacteria. Bacterial cells were grown in LB medium overnight at 28 °C and adjusted to an OD_600_ of 0.8. The antagonistic activity of *Bacillus* strains against *C. vitis* GP1 was assessed through plate bioassays as described by Wang [41]. *G. fructigrum*, *B. cinerea*, *D. eres*, *A. viticola*, *F. oxysporum*, *A. niger*, and *P. clavispora*, which were kept in our library, were used to determine the percentage inhibition of mycelial growth of other fungal pathogens in grapes. The fungal pathogens were cultured on potato dextrose agar (PDA) plates (20 g glucose, 200 g potato, 15 g agar, 1000 mL distilled water, pH 7). Additionally, the antagonistic activity of GSBZ09 against *Allorhizobium vitis* was determined using the double-layer agar plate method (15 mL bottom-layer PDA, 10 mL top-layer 2% agar) [42]. The percentage of growth inhibition was calculated by the following equation: n = [(A − B)/A] × 100, where A is the colony diameter of control fungi and B is the colony diameter of treated fungi. The values were recorded as the means of four replicates, and each experiment was repeated three times.

### 4.2. Identification of Strain GSBZ09

#### 4.2.1. Genomic DNA Extraction and Phylogenetic Analysis

The strain GSBZ09 was cultured in LB media at 28 °C with shaking at 200 rpm for 12 h. Genomic DNA was extracted from cultured GSBZ09 cells (OD_600_ = 0.8) using a TIANamp Bacteria DNA kit (Tiangen Biotech Co., Ltd., Beijing, China). The taxonomic position of strain GSBZ09 was determined by multilocus gene sequence analysis (MLSA) based on six housekeeping genes (16S rRNA, *gyrB*, *rpoD*, *atpD*, *rho* and *pgk*). The phylogenetic tree was constructed using the maximum likelihood method in MEGA 6.0. Other available gene sequences of closely related species for phylogenetic tree construction were downloaded from the NCBI database (Appendix A).

#### 4.2.2. Morphological, Physiological and Biochemical Tests

Strain GSBZ09 was cultured on LB medium plates and incubated at 28 °C for 24 h to observe the colony characteristics, such as the color, morphology and growth properties. The strain morphologies were observed by scanning electron microscopy (SEM) and transmission electron microscopy (TEM 1230 microscope, JEOL) [43]. Physiological and biochemical tests were performed using the Biolog GN2 microplate system (Biolog, Haywood, CA, USA) [44]. The growth curve and the dynamic change in pH were measured every 4 h.

In vitro characterization of strain GSBZ09 was checked for growth at diverse pH, salt concentration and temperature. To check the pH sensitivity, the strain GBSZ09 was cultured at 28 °C for 24 h in tubes containing 5 mL of LB medium maintained at 3, 5, 7, 9, 11, 13 pH. Similarly, the strain GSBZ09 was grown in LB media tubes containing different salt concentration (1, 2, 3.5, 5, 6.5, 8%,) to observe its sensitivity to salt. For temperature tests, the GBSZ09 was cultured in LB media tubes, which incubated at different temperature (10, 30, 50, 70, 80, 90, 100 °C).

### 4.3. Measurement of Extracellular Enzyme Production

#### 4.3.1. Protease Production

Skim milk agar plates (skim milk, 100 g; peptone, 5 g; agar, 15 g; distilled water, 1000 mL) were prepared to detect Protease production. *B. velezensis* GSBZ09 overnight at 28 °C with shaking at 200 rpm, then 2 µL of GSBZ09 culture (OD_600_ = 0.8) was inoculated onto the center of skim milk agar plates. After 48 h of incubation at 28 °C, protease activity was observed based on the presence of clear zones around bacterial cultures.

#### 4.3.2. Cellulose Degradation

The cellulose-degrading ability of *B. velezensis* GSBZ09 (OD_600_ = 0.8) was determined by the medium (carboxymethyl cellulose (CMC, 1 g; Na_3_PO_4_, 25 mmol; pH 7.0; agar, 20 g; and distilled water, 1000 mL) [41]. After incubation at 28 °C for 5 d, the plates were rinsed for 15 min with a 1% solution of Congo red dye and then washed twice with 1 M NaCl buffer. The diameter of the clear zone around the colony (cellulose degradation) was measured [45].

#### 4.3.3. Amylase Production

The ability of *B. velezensis* GSBZ09 to produce amylase was determined in medium containing soluble starch 5 g/L, NaCl 5 g/L, and agar 20 g/L (Lugol solution; MERCK). After 48 h of incubation at 28 °C, the plates were stained with an iodine solution.

#### 4.3.4. Lipase Production

Selective medium containing 10 g of peptone, 0.1 g of CaCl_2_, 2.5 g of NaCl, 10 mL of Tween 20, and 15 g of agar was used to determine the ability of *B. velezensis* GSBZ09 to produce lipase. The bacteria were streaked on the medium and incubated at 28 °C for 72 h. Lipase activity was determined based on the presence of depositions around bacterial colonies [46].

### 4.4. Measurement of IAA Production, Siderophores and Mineral Phosphate Solubilization

Strain GSBZ09 was cultured in DF (peptone, 5.0 g; yeast extract, 1.5 g; beef extract, 1.5 g/L; NaCl, 5.0 g/L; tryptophan, 0.5 g/L) salt minimal medium, and L-tryptophan was added to the medium to a concentration of 1.02 g/L [47]. After incubation for 24 h at 28 °C, the IAA concentration was estimated according to the method described by Yuan et al. (2011) [48]. National Botanical Research Institute Phosphate (NBRIP) solid medium was used to determine the capability of strain GSBZ09 to solubilize phosphate [49]. Ten microliters of bacterial culture were dropped on sterile filter paper (diameter, 5 mm) and placed in the middle of agar plates containing NBRIP. The clear zone around the colony was measured after 7 d at 28 °C [50]. A CAS agar plate was used for qualitative analysis of siderophores, and yellow circles that appeared around the colonies were measured after 7 d of incubation at 28 °C [51].

### 4.5. Antibiotic Resistance and Hemolysis Assay

Nine antibiotics, spectinomycin (50 µg/mL), ampicillin (200 µg/mL), vancomycin (50 µg/mL), kanamycin (50 µg/mL), streptomycin (10 µg/mL), gentamycin (10 µg/mL), chloramphenicol (20 µg/mL), tetracycline (5 µg/mL) and rifampicin (10 µg/mL), were used to test the characteristics of the antibiotic resistance of GSBZ09. The MIC and MBC of ampicillin for strain GSBZ09 were determined as previously described [52].

The strain GSBZ09 was cultured in LB media at 28 °C with shaking at 200 rpm for 12 h, and 5 μL of bacterial culture was dropped on Wagstsuma Blood Agar Base (Hopebio, Qingdao, China) and then cultured at 28 °C. Hemolysis was determined by observation of a cleared zone surrounding bacteria grown on the plates [53].

### 4.6. Assessment of Biocontrol Activity and Plant Growth Promotion

To evaluate the plant growth promotion activity, five young 1-year-old plants of *V. vinifera* cv. RG were selected for treatment with 50 mL of GSBZ09 culture (10^8^ CFU/mL) and culture filtrate by irrigation, irrigation treatment were conducted weekly during a month. Additionally, another five plants used as controls were treated with sterile water. The weight of the root and shoot of cv. RG were measured 60 d after the irrigation. The SOD, PPO and PAL enzyme activities in the leaves were detected by previously described methods [54]. In addition, the MDA, protein and proline contents of the leaves were measured using previously published methods [55,56,57].

The pathogenic strain *C. vitis* GP1 of grape white rot was grown on PDA plates for 3 d at 28 °C. The strain was used to inoculate *V. vinifera* cv. RG by detached leaves and fruit inoculation. Surface sterilization of the leaves and fruits was performed as previously described [3]. PDA plugs (8 mm) with fungi were inoculated on the leaves, and fruit were inoculated with 20 μL of spore suspension (1 × 10^7^ spores/mL). For the control, we used PDA plugs without fungi and sterilized water. One day after inoculation, the GSBZ09 cultures at a concentration of 10^8^ CFU/mL were sprayed to test their ability to control grape white rot. GSBZ09 cultures sprayed before inoculation of the pathogenic strain *C. vitis* GP1 were used to test the prevention effect. All the leaves and fruit were maintained at 28 °C and 80% to 90% RH. The incidence, lesion diameter of the leaves, disease index of the fruit, and control efficiency of the GSBZ09 cultures and the culture filtrate were recorded 5 d after inoculation.

### 4.7. Statistical Analysis

The data were analyzed based on an ANOVA and Duncan’s multiple range test (*p* ≤ 0.05) using the statistical software SPSS version 19.0 (SPSS Inc., Chicago, IL, USA). The significance between the means of different treatments was evaluated using Duncan’s (D) test.

## 5. Conclusions

*B. velezensis* GSBZ09 isolated from grapevine had a broad range of antagonistic activities against many fungal and bacterial pathogens on grape. Furthermore, GSBZ09 showed a variety of beneficial features, including IAA production, phosphate solubilization and siderophores, thus revealing its beneficial features related to plant growth promotion. Further study showed that *B. velezensis* GSBZ09 could produce cellulase, protease, amylase and lipase and improve the level of defense-related antioxidant enzymes in grape. Additionally, GSBZ09 culture and culture filtrate exhibited significant biological control efficacy on grape write rot. All these features indicated that strain GSBZ09 could be a promising biocontrol agent for grape disease control and could promote the practical application of strain GSBZ09.

## Figures and Tables

**Figure 1 pathogens-11-00248-f001:**
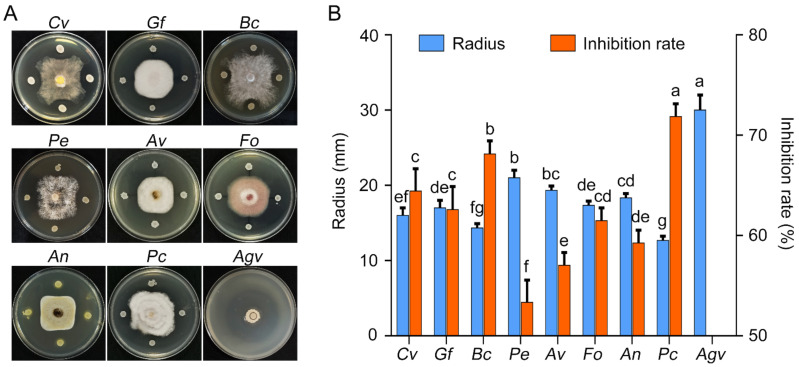
Antagonistic assay of *Bacillus velezensis* GSBZ09 against eight pathogenic fungi and one pathogenic bacterium. (**A**) Antagonistic assay of *B. velezensis* GSBZ09. *Coniella vitis* (CV). *Gloeosporium fructigrum* (GF). *Botrytis cinerea* (BC)*. Diaporthe eres* (Pe). *Alternaria viticola* (Av). *Fusarium oxysporum* (Fo). *Aspergillus niger* (An). *Pestalotiopsis clavispora* (Pc). *Allorhizobium vitis* (Agv). (**B**) Colony radius and inhibition rate of each microorganism. Error bars represents the means ± standard deviation of three replicate experiments. Different letters above the bars indicate a significant difference at *p* < 0.05 according to Duncan’s multiple-range test.

**Figure 2 pathogens-11-00248-f002:**
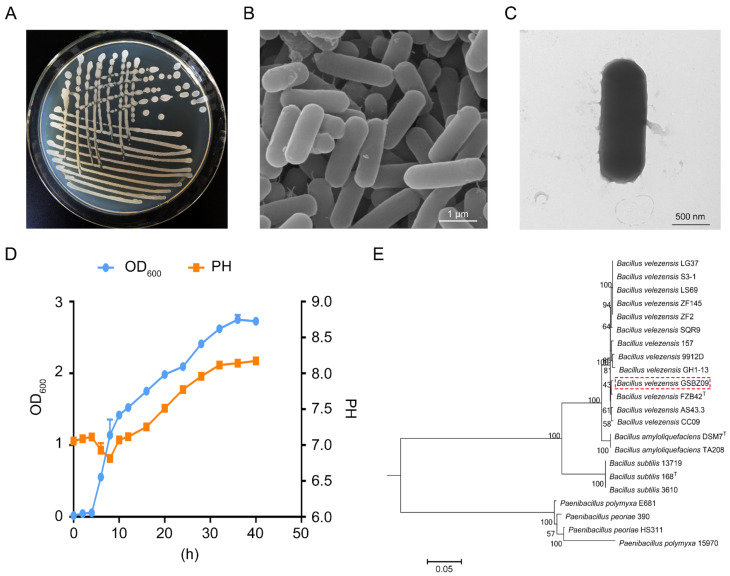
General characteristics of *Bacillus velezensis* GSBZ09. (**A**) Image of GSBZ09 colony morphology. Image of GSBZ09 cells using scanning electron microscopy (**B**) and transmission electron microscopy (**C**). (**D**) Phylogenetic tree of *B. velezensis* GSBZ09 among other *Bacillus* species. The phylogenetic tree was constructed based on six housekeeping genes (16S rRNA, *gyrB*, *rpoD*, *atpD*, *rho*, and *pgk*) according to the aligned gene sequences using the maximum likelihood method in MEGA 6.0. Bootstrap values (1000 replicates) are shown at the branch points. The scale bar indicates 0.05 nucleotide substitutions per nucleotide position. (**E**) Growth dynamics and pH change of *B. velezensis* GSBZ09. Error bars represents the means ± standard deviation of three replicate experiments.

**Figure 3 pathogens-11-00248-f003:**
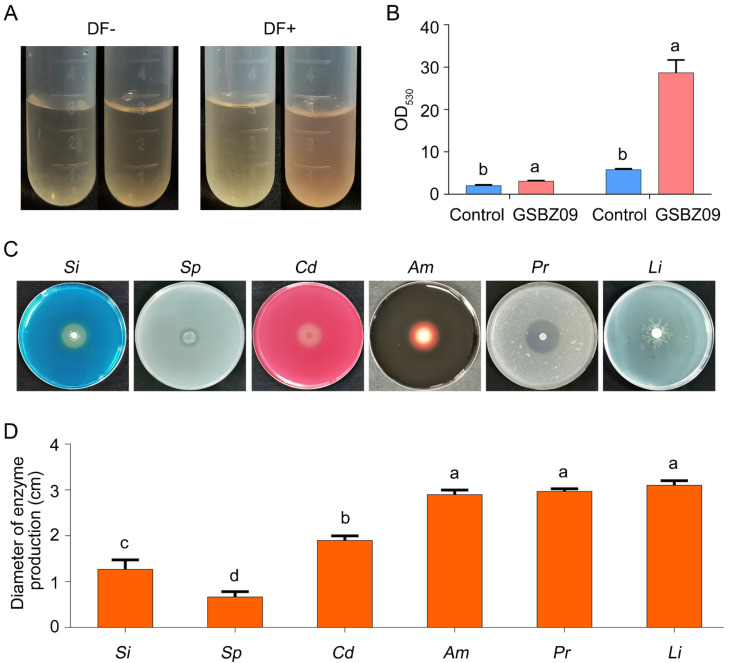
Detection of extracellular enzyme production and growth-promoting traits of *Bacillus velezensis* GSBZ09. (**A**) IAA production of *B. velezensis* GSBZ09. (**B**) Determination of indole-3-acetic acid (IAA). (**C**) mineral phosphate solubilization (Sp), siderophores (Si), cellulose degradation (Cd), amylase (Am), protease (Pr), and lipase (Li). (**D**) Diameter of the zone produced on the plates. Error bars represents the means ± standard deviation of three replicate experiments. Different letters above the bars indicate a significant difference at *p* < 0.05 according to Duncan’s multi-range test.

**Figure 4 pathogens-11-00248-f004:**
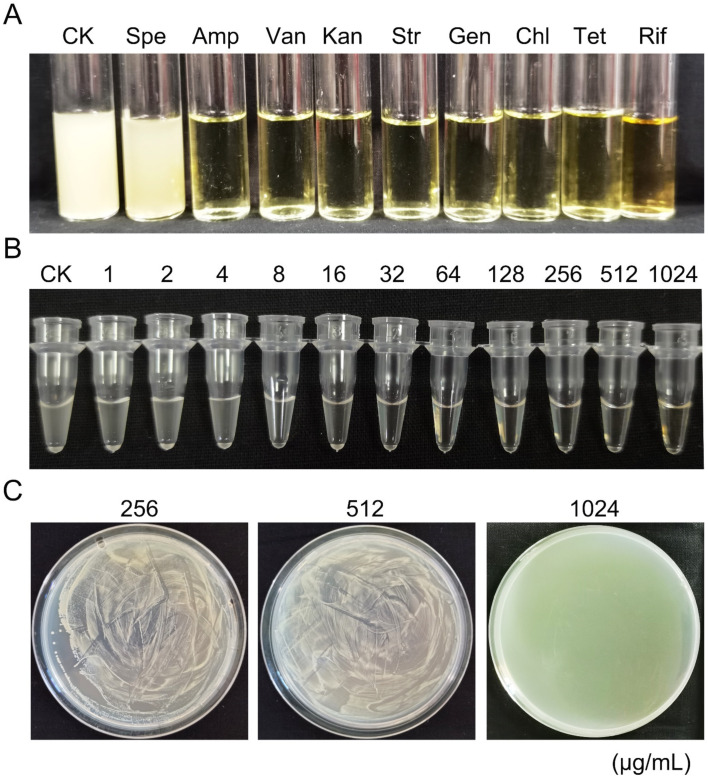
Determination of antibiotic resistance of *Bacillus. velezensis* GSBZ09. (**A**) Survival of *B. velezensis* GSBZ09 treated with different antibiotics. Spectinomycin (Spe), ampicillin (Amp), vancomycin (Van), kanamycin (Kan), streptomycin (Str), gentamycin (Gen), chloramphenicol (Chl), tetracycline (Tet) and rifampicin (Rif). (**B**) Minimum inhibitory concentration (MIC) of spectinomycin for strain GSBZ09. (**C**) Minimum bactericidal concentration (MBC) of spectinomycin for strain GSBZ09.

**Figure 5 pathogens-11-00248-f005:**
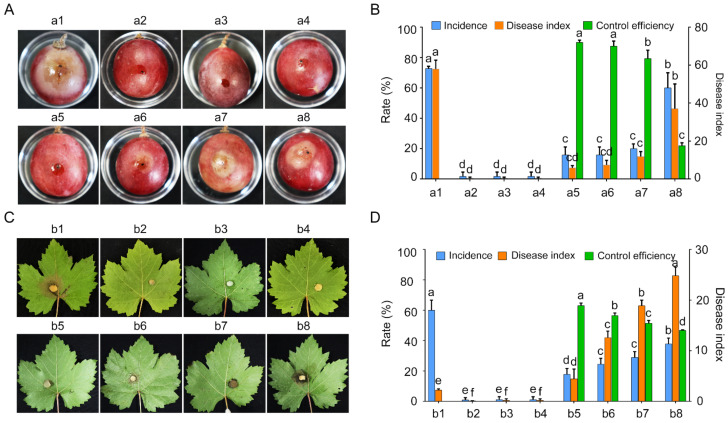
Biocontrol efficiency of *Bacillus velezensis* GSBZ09 on grape white rot caused by *Coniella vitis.* (**A**,**C**) (a1,b1) Inoculated with *C. vitis*; (a2,b2) LB broth; (a3,b3) culture of GSBZ09, (a4,b4) culture filtrate of GSBZ09; inoculated with *C. vitis* 24 h after inoculation with the culture (a5, b5) and culture filtrate (a6, b6) of GSBZ09; inoculated culture (a7,b7) and culture filtrate (a8,b8) of GSBZ09 24 h after inoculation with *C. vitis.* (**B**,**D**) Incidence, disease index and control efficiency of *B. velezensis* GSBZ09. Error bars represent the means ± standard deviation. Ten biological replicates were performed for each treatment, and the experiments were independently repeated three times. Different letters above the bars indicate a significant difference at *p* < 0.05 according to Duncan’s multi-range test.

**Figure 6 pathogens-11-00248-f006:**
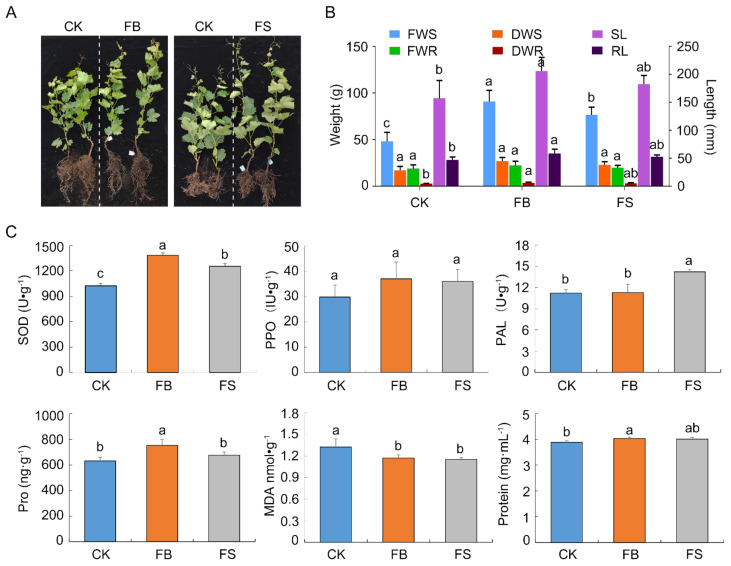
Growth-promoting effect of *B. velezensis* GSBZ09 on grape. (**A**) Aspect of grape treatment with LB broth (CK) culture (FB) and culture filtrate (FS) of GSBZ09. (**B**) Fresh weight of shoots (FWS), dry weight of shoots (DWS), fresh weight of roots (FWT), dry weight of roots (DWT), shoot length (SL), and root length (RL) of grapes. (**C**) Superoxide dismutase (SOD), polyphenol oxidase (PPO) and phenylalanine ammonia (PAL) enzyme activities in the leaves; malondialdehyde (MDA), protein (Protein) and proline (Pro) of the leaves. Error bars represents the means ± standard deviation of five replicate experiments. Different letters above the bars indicate a significant difference at *p* < 0.05 according to Duncan’s multi-range test.

## Data Availability

Not applicable.

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
