# Peer review of "Suppression of Grape White Rot Caused by Coniella vitis Using the Potential Biocontrol Agent Bacillus velezensis GSBZ09"

_pathogens, 2022, doi:10.3390/pathogens11020248_

Round 1
Reviewer 1 Report
Comments and Suggestions for Authors
The paper is interesting. However to improve publication quality I have the following suggestions:
- Results shoud be corrected; especially check the order of figures and attached captions and descriptions in the text. Additionally, this section should be supplemented with the results of statistical tests.
- Materials and methods should be improved by adding some information to make more clear all the experimentation e.g. detail of pot experiment.
Line 30.
barriers ?
Line 72-78
Develop the aim of the work. Indicate the innovative aspect of this work
Line 76-78
This sentence should be included in the summary of the results.
Line 92
antifungal and antibacterial spectrum
Line 101
Colony radius and inhibition rate of each microorganisms.
Line 108-109
There is no information about the source of the presented results; no data in the materials and methods regarding the measurement of biomass growth in a differ temperature, salinity or pH range.
Line 110
(hpi)? What is mean?
Line 109-112
This description relates to Figure 2E.
Line 115-119
This description does not apply to Figure 2E.
Line 129-135
This description does not apply to Figure 3.
Line 137-140
The description does not apply to the attached figure.
Line 145-146
The description does not apply to the attached figure.
Line 151-155
The description does not apply to the attached figure.
Line 169-173
Where are these results shown in the form of a figure or a table?
Line 175-179
This description does not apply to the attached figure.
Line 189
The description does not apply to Figure 6
Line 199 -203
This description does not apply to the attached figure.
Line 257-258
Explain what this sentence is about: „MDA can affect plant growth and development by damaging cellular biofilms”
Line 245-263
The increase in the activity of antioxidant enzymes can also be interpreted as a response to oxidative stress.
Line 277
Remove „agar”
Line 302
Remove „B. velezensis GSBZ09 overnight at 28℃”
Line 310
Incubation lasted 5 days or 48 hours? Clarify.
L 349
What does regular irrigation mean? Clarify
Line 367 -370
In this work, I did not find the results of the indicated statistical tests. Please place them in the Results section.
Line 382
Bacillus
Suplementary Materials
There is no information about Table S1 and Table S2.
Reviewer 2 Report
Reviewer 1:
Comments to the Author
General comments: The manuscript by Yin et al., (submitted) describes the suppression of grape white rot caused by Coniella vitis by Bacillus velezensis GSBZ09. This is an interesting study which shows the potential of a Bacillus strain isolated from a vineyard soil, to protect this disease. The strain was well characterized morphologically, taxonomically and biochemically. Strain GSBZ09 reduced disease index of grape white rot and promoted the growth of grapevine plants. However, there were several significant mistakes in the text, legends and figures which should be corrected. For this manuscript, kindly address specific comments below.
Specific comments
Abstract Section:
Line 1: Choose article type
Lines 9-10: Better to rephrase this sentence “Bacillus velezensis is a reputable plant growth-promoting bacteria.”
Line 16: Replace “At the same time” with “Furthermore”.
Lines 18-20: Rephrase sentence “Pot experiments showed that the application of GSBZ09 significantly decreased disease index of the grape white rot, directly promoted the growth of grapes and upregulated defense-related enzymes”.
Line 21: Replace “highly potent and competitive” with “potential”.
Main Text:
Line 27: Rephrase “…is considered to be the main pathogen of the disease….”
Line 30: delete “The” which is before “infected”.
Line 40: Replace “it can grow” with “these species can grow”
Line 48-49: Rephrase sentence “B. velezensis group strains are reputed to be PGPR”.
Line 49: Rephrase “The production of IAA and cytokinin, both of which are …….”
Line 57: Rephrase “….resveratrol in grapes and further reduce the occurrence….”
Line 66-68: The pathogens mentioned here are not bacteria! So the mention of “antibacterial activity” is incorrect. Plasmopara is an oomycete while the others are fungi. Correct this!
Line 76: I find it weird that you use the word “better” here. I guess you mean that the strain is better when compared to other isolates? Then you should mention it here, otherwise, you will need to remove this word.
Line 85: Is Figure S2B referring to 1% culture filtrate and S2C referring to 20% culture filtrate? It does not seem correct when you compare these pictures with your growth inhibition. The growth inhibition rate of B seems better than that of C. Please cross check your data again.
Line 91: Rephrase: ….both the center of the germ tube and the spores swelled, and the contents leaked…..
Line 92: same comment as Lines 66-68. These pathogens are not bacteria!!!
Line 112: It should be Figure 2E and not Figure 2D
Line 119: It should be Figure 2D and not Figure 2E
Line 121: Rearrange the figures such that “Growth dynamics and pH change figure comes first (2E)” before the phylogenetic tree (2D)” since this is the order in which you discussed your results.
Line 130: which plate are you referring to?
Lines 133-134: Rephrase: “….to fix nitrogen. This strain can produce IAA and siderophores, which indicates its high potential as a competitive strain….”
Lines 136-140: You interchanged legend of Figure 4 with Figure 3 so this legend is for the wrong figure. This is really confusing. Please correct this.
Lines 151-155: Please correct the figure and legend. See comment of Lines 136-140.
Line 164: Rephrase: “…One treatment consisted of grape inoculation with GSBZ09….”
Line 165: Rephrase: “….to determine the preventive effect,…”
Lines 175-179: Again, you have interchanged Figure 5 and Figure 6. This Figure is not Figure 5 but Figure 6.
Lines 198-203: This Figure should be Figure 5.
Line 209: Again, it is not possible to have antibacterial activity against fungal pathogens!
Line 226: delete “of”
Line 227: replace “for plants” with “by plants”
Line 232: Replace “the results” with “our results”
Line 248: replace “might” with “may”
Lines 251-252: Rephrase: “…, which may have resulted in the observed morphological changes….”
Line 274: Delete “To determine”. Start sentence with “The percentage inhibition …”
Line 276: Replace “were detected” with “was determined”.
Line 277: Be specific about the quantity of agar used: 15g or 20g?
Line 280: What does “..top-layer 2 agar” mean?
Lines 302-306: What quantity of bacterial culture was inoculated into wells? Indicate this.
Lines 314-317: What quantity or concentration of bacteria was inoculated? Was it an overnight culture?
Lines 416-417: Italicize genus and specie names and capitalize first letter of Genus names
Line 426-427: Same reference is repeated in Line 461-462.
Line 453: italicize Eucommia ulmoides
Line 477: italicize streptococcal endocarditis and capitalize the first letter of the genus name.
Line 510: italicize Valeriana officinalis
Line 522: italicize Escherichia coli
Reviewer 3 Report
The present work uses bacterial isolates isolated from vineyard soil to control grape white rot caused by Coniella vitis. The authors have done a lot of work and results seem promising in finding a biocontrol agent to control the studied disease. Overall, I want to congratulate authors for their interesting work.
Despite that i have some questions and suggestions that can be found below:
Lines 33-34: I believe authors mean that these Bacillus species have been identified as PGPR.
Lines 33-43: I suggest putting this paragraph after paragraph in lines 44-53 or rearrange information on these paragraphs. This because there is no reason for describing Bacillus species in the second paragraph and, for the benefit of train of thought, it would be better to introduce PGPR first.
Results in 2.1: Were results statically significant in relation to control? I don’t see the statistics anywhere in the text manuscript or figures. Also, results for IAA, diameter of enzyme production should have the statistical results presented. I can see * in figure 5 (that should be figure 6) but I don’t see explanation for that in the figure legend.
Line 112: Authors are referring to figure 2E and not 2D. Please change order of figure or text.
Figures: I believe figure 3 and 4 are misplaced. Figures 5 and 6 are also misplaced and legend of figure 5 is incomplete. In figure 6 legend replace “(Protin)” with “(Protein)”
Lines 189-191: I see statistical differences in MDA in relation with the control but this sentence leads to believe that there are none. Also, authors state that there are no significant differences in proline levels but the graph shows that there are in the case of FB.
Line 209: I believe that authors mean antifungal activity
Line 266: which indicators did authors take into account to conclude that sampled soil was healthy? Or do authors mean that vineyards from which soil was collected were apparently free of disease? Also, how was soil collected? Was it randomly collected in the vineyard? Considering that “ten grams of soil were weighted”, I believe that may have collected soil near the roots and not “rhizosphere soil” because to obtain rhizosphere soil, authors should have collected roots and use soil that was adjacent to roots. Please clarify this part of the methodology.
Line 267: replace “was” with “were”
Line 271: what does “purified for further investigation” means? Do authors mean that pure cultures were used?
Line 276: I believe authors mean that these fungal pathogens “were used” instead of “were detected”. Or did authors also isolate them from collected soil samples? Where did authors obtained these fungal isolates?
Lines 265-284: Please clarify how B. velezensis was identified? Did authors used sequencing of 16S region? In line 271 authors state that Bacillus like strains were selected and from then on there is no reference of how was this species identified. Reading the results, I can see that this was selected based on the ability to inhibit different fungal pathogens but that is not clear in material and method section. Please try to reformulate 4.1 sub-chapter to explain that more clearly. Also, I suggest either not naming GSBZ09 before the identification section or changing the order (4.2.2 to be 4.2.1) improve reading.
Lines 289-290: Please specify the used methods
Line 302: I am not sure the sentence “B. velezensis GSBZ09 overnight at 28℃” belongs in here.
Line 304: Please clarify the inoculation into wells. Were wells made in the agar plates?
Line 309: Please substitute “by the medium (carboxymethyl cellulose (CMC), 1 g;” with “by using a medium with carboxymethyl cellulose (CMC, 1 g;”
Lines 348-350: Were culture filtrates prepared separately? For how long were plants irrigated with culture and filtrates?
Round 2
Reviewer 1 Report
Dear Authors,
Thank you for comprehensive answers. The work has been thoroughly revised. I suggest to accept this manuscript.
Reviewer 2 Report
Dear authors,
Thank you for correcting this manuscript.
There is need for a slight correction:
Line 11: change "bacterial" to "bacteria"